# Do You Want to Increase Physical Activity in Adolescents? A School-Based Physical Activity Program Could Be an Efficient Way

**DOI:** 10.3390/children10101641

**Published:** 2023-09-30

**Authors:** Beatriz Polo-Recuero, Alfonso Ordóñez-Dios, Miguel Ángel Rojo-Tirado, Alberto Lorenzo

**Affiliations:** 1Department of Sports, Faculty of Physical Activity and Sport Sciences, Universidad Politécnica de Madrid, 28040 Madrid, Spain; bpolo@educa.madrid.org; 2Faculty of Health Sciences, Universidad Rey Juan Carlos, 28933 Madrid, Spain; 3Department of Physical Education, Sport and Human Motor Skills, Faculty of Teacher Training and Education, Universidad Autónoma de Madrid, 28049 Madrid, Spain; afordonez@educa.madrid.org; 4LFE Research Group, Department of Health and Human Performance, Faculty of Physical Activity and Sport Sciences, Universidad Politécnica de Madrid, 28040 Madrid, Spain; ma.rojo@upm.es

**Keywords:** physically active lessons, bike desks, sedentarism, academic performance, physical performance, physical activity

## Abstract

The aim of this study was to assess the effects of a classroom-based physical activity program, using bike desks, on academic and physical performance in adolescents. The Program to Enhance and Develop Active Lessons (PEDAL) was designed for this purpose, expecting an increase in students’ physical activity without any decrease in academic performance. This intervention based on pedal or bike desks—stationary bikes that integrate with a desk workspace—was conducted with 55 high- school students who were randomly assigned to two groups: a PEDAL group (*n* = 28, 14.86 ± 0.65 years old, 46.4% girls) and a control group (*n* = 27, 15 ± 0.68 years old, 51.9% girls). Throughout the intervention, the PEDAL students pedaled 4 days a week for 10 weeks during their Spanish-language arts lessons. The comparisons between the PEDAL group and the control group, as well as the pre- and post-test results, were statistically analyzed to verify the students’ physical activity (i.e., IPAQ-SF, heart rate monitors, polar OH1^+^), cardiorespiratory capacity (20 m shuttle run test), and academic performance (d2 test of attention and language proficiency test). Regarding the physical aspect, only the PEDAL group showed significant growth in their physical activity levels as compared to the pre-test data (*p* = 0.001), and they achieved higher results compared with the control group (*p* = 0.022) and less sedentary time than control students (*p* = 0.012). Concerning cardiorespiratory fitness, there were no post-test differences between the two groups (*p* = 0.697), probably because the physical activity performed with the bike desks was light–moderate. As far as academic performance is concerned, no significant post-test effects were discovered in either group on the levels of language competence (*p* = 0.48), attention (TOT, *p* = 0.432), and concentration (CON, *p* = 0.216). In conclusion, adolescents who move while learning, using bike desks, increase their light and moderate physical activity without any detriment to academic performance.

## 1. Introduction

Globally, the levels of overweight and obesity in children and adolescents are showing a rising trend [1], and an active life is a powerful public health strategy to revert this negative pattern [2]. Insufficient physical activity and increased amounts of sedentary time are associated with health risks [3]. The Global Action Plan for the Prevention and Control of Noncommunicable Diseases has called to reduce inactivity by 10% before 2025 (WHO 2010). There are several strategies that could be proposed to increase adolescents’ physical activity, and schools seem to be the ideal place to reach every single student [4] and increase their physical activity time [5]. One of the aims of school-based physical activity programs is to provide children and youth with more physical activity and less sedentary time. In the context of school, there are different ways to promote active lifestyles, such as active recess (i.e., providing students with time and resources to be active during recess), active breaks (i.e., short physical activity breaks within a lesson or between lessons), physically active learning (i.e., creating learning through movement), sport tournaments, or more physical education lessons, hereinafter referred to as PE [6]. These interventions have emerged to target inactivity, obesity, and sedentarism.

This paper compiles the physical and academic outcomes discovered after the implementation of the Program to Enhance and Develop Active Lessons (PEDAL), a physical activity intervention. This program is based on physically active learning (PAL), an opportunity for learning combined with movement within the school day [7]. There are two ways to implement PAL: changing the lessons’ methodology [8,9] or integrating active desks/workstations into the classroom [10,11].

Active desks can also be called active workstations, fit desks, or dynamic seating devices, and they are a way to learn academic content while moving. There are different types of active desks, including standing desks, balance desks, stepper desks, strider desks, treadmill desks, fitballs, wobble chairs, and bike desks. They all have in common a strategy to integrate different devices into the traditional school desk to practice physical activity.

The PEDAL program analyzed the effects of bike desks during Spanish-language arts lessons. Only a few studies have implemented bike desks for secondary students in a real education context; most of them were conducted in the US [12]. Concerning Europe, only one study has been put into action [10], and, as far as is known, PEDAL is a pioneer, establishing bike desks in a school in Spain. The majority of the results of the available studies describe the bike desk strategy as feasible and/or acceptable [13,14], with positive effects on physical activity [12]. However, it is not only the physical benefits that are important; this research is crucial in order to continue to investigate whether PAL strategies, and bike desks in particular, could pose difficultiesfor students’ learning, although it has been found that moderate physical activity has beneficial effects on cognitive function and concentration [15,16]. Regarding bike desks, only one study has examined their influence on academic outcomes [10], while several studies have controlled on-task behavior [13,17] Therefore, the main goal of the present research was to measure the effect of an educational program (PEDAL), based on the use of active desks (bike desks) during school time, on students’ physical activity, physical condition, and academic performance. Based on a systematic review that was conducted [18], it seems that the PEDAL program could help in achieving the physical activity goal set by the WHO, and it is hypothesized that less active students would benefit more from the intervention than those with high physical activity levels, and that gender differences could emerge.

## 2. Materials and Methods

### 2.1. Participants and Study Design

A total of 60 students in their fourth year of secondary education were eligible for participation in the study. Due to the lack of financial support to provide PEDAL with more bike desks, and the organizational requirements of the school, the program could only be implemented in two different classes. In this regard, it should be noted that previous experimental studies on bicycle desks with students have had similar sample sizes, with an average of 52.83 students [18]. Finally, 60 families agreed to participate (participation rate: 100%) by signing an informed consent form. However, five students were excluded (one participant left the study voluntarily, one suffered a long-term injury, and three did not meet the 80% minimum class attendance requirement). As a result, 55 students (28 boys and 27 girls) from a public high school in Madrid participated in the study and were randomly divided into two groups: a PEDAL group (*n* = 28, 14.86 ± 0.65 years old, 46.4% girls) and a control group (*n* = 27, 15 ± 0.68 years old, 51.9% girls). Data collection took place from October 2019 to December 2019. The study protocol passed the Review Committee for Research Involving Human Subjects evaluation from a renowned university and adhered to the Helsinki Declaration of 1961, revised in Fortaleza (Brazil) in 2013.

### 2.2. Procedure

Two classes participated in the study, both being randomly divided into two groups as previously mentioned. Both classes used an active-desk-enabled classroom during their language arts courses, equipped with 15 DeskCycle2^TM^ (3D Innovations, LLC., Greeley, CO, USA) bike desks. Neither the PEDAL nor the control group changed their usual language arts lessons in terms of dynamics and tasks; the only difference resided in the voluntary pedaling for PEDAL group students during the aforementioned lessons. They could choose individually the moments throughout the lesson to pedal, while listening, speaking, reading, or writing. Students in this intervention group collected their time and distance data as well as the resistance applied during the active lessons and recorded themin their logs. Moreover, within each group, 18 participants were asked to wear an HR monitor, the Polar OH1^+^ (Polar Electro Oy, Kempele, Finland), throughout the entire school day for five academic days of the week, during four different assessment periods (pre-test, week 1, week 4, and week 7).

The school day ran from 8:15 a.m. to 2:10 p.m. and was divided into 55-min blocks for each subject (i.e., math, language arts, science). In total, all the students, in both the control and the PEDAL groups, had 220 min of language arts per week.

The PEDAL program was applied in 35 active lessons and it lasted 10 weeks. Before starting the study (familiarization stage) and during the 10th week of the intervention period, students were physically evaluated with a 20 m shuttle run test [19], a language competence test [20], and an Attention Concentration test [21]. Additionally, the level of physical activity was assessed using the International Physical Activity Questionnaire (IPAQ) [22]. Finally, students’ heart rate (HR) follow-ups were carried out every three weeks using pulsometers.

### 2.3. Instruments

Cardiorespiratory fitness. This variable was assessed using a 20 m shuttle run test [19] in which students had to run back and forth between 2 lines that were 20 m apart following an audio signal. The test ended when participants failed to reach the end lines concurrent with the audio signal on 2 consecutive occasions or when the participant stopped because of exhaustion. For higher sensitivity, the test results were expressed as the number of stages completed, instead of in minutes, as is habitually done.

Physical activity. The level of physical activity was measured using the International Physical Activity Questionnaire Short Form (IPAQ-SF) [22]. This questionnaire assesses the types of intensity of physical activity and sedentary time that students engage in as part of their daily lives, based on the last 7 days. According to IPAQ-SF, results are shown as grades of physical activity performed (1: low, 2: moderate, 3: high), metabolic equivalents (MET-minute/week), and sedentary time (minutes/week). Furthermore, the PEDAL students were categorized according to their pre-test physical activity (MET-minute/week) in order to analyze the effect of pedaling according to their starting point, as previously described in the literature [23]. The IPAQ scoring protocol categorizes the subjects’ physical activity into three categories: low (<600 MET), moderate (600–3000 MET), and high (>3000 MET).

Language competence. This was measured and evaluated using a regional test. The Community of Madrid (a region of Spain) has been using this test to assess secondary-school last-year students in language, math, and social and civic competence for several years. An official handbook to apply and correct this test is available (Resolución de 8 de marzo de 2019).

Attention (TOT) and concentration (CON). These abilities were measured using the d2 Test of Attention [21]. The participants must discern and cross out the letter “d” during a certain and controlled period of time. The TOT and CON data were analyzed to clarify attention and concentration, respectively.

Heart rate (HR). The monitoring of HR was used to analyze the intensity of the physical activity performed. It is an objective method and, along with accelerometers, is the best way to detect and evaluate physical activity patterns in children and adolescents [24]. The difficulties in education to individually calibrate each subject led to establishing certain HR threshold points [25]. According to them, for adolescents from 11 to 16 years of age, when the HR is over 140 beats per minute for at least 10 min, they are engaged in moderate physical activity. Data were stored and analyzed in the PolarFlow application.

Prior to the implementation of the PEDAL program, during the 2018–2019 school year, a pilot project was carried out in order to help students to become familiar with some of the instruments.

### 2.4. Data Analysis

Statistical analyses were conducted using the SPSS software (SPSS v.22, IBM Corporation, New York, USA). The significance level was set at *p* < 0.05. The descriptive characteristics of the sample were calculated. Normality was checked using the Kolmogorov–Smirnov test. The main analysis searched for differences in the following dependent variables: pedaling characteristics (i.e., resistance, time, and distance), physical conditions, language competences, and attention abilities (effectiveness and concentration), as well as analyzed physical activity variables (MET-minute/week and sedentary time) and HR, using an analysis of variance (ANOVA) of repeated measures. We compared the data based on the independent variables intervention group (PEDAL or control) and gender (boy or girl). Additionally, cardiorespiratory fitness, language competence, attention (effectiveness and concentration), physical activity, and changes in the amount of sedentary time were analyzed depending on their initial physical activity groups by an analysis of variance (ANOVA) of repeated measures, for the PEDAL group. The effect size was calculated using the partial eta-squared (ηp^2^) ratio of variance, and small, moderate, and large effects corresponded to values equal to or greater than 0.10, 0.25, and 0.40 [26].

## 3. Results

### 3.1. Results Data

#### 3.1.1. Participants’ Characteristics and Pedaling

In every active lesson, students pedaled a mean of 33.82 ± 1.13 min, covering 13.12 ± 0.87 km (8.15 ± 0.54 miles), with a resistance of 2.84 ± 0.0.10. There were no significant differences between boys and girls in the PEDAL group in terms of the daily mean pedal time (*p* = 0.892), distance accomplished (*p* = 0.928), and applied resistance (*p* = 1.000).

The main effect of moment was observed for all pedaling assessments: resistance (Wilks’ λ = 0.173; ηp^2^ = 0.827; F_2,24_ = 57.404; *p* < 0.001), time (Wilks’ λ = 0.428; ηp^2^ = 0.572; F_2,24_ = 16.010; *p* < 0.001), and distance (Wilks’ λ = 0.245; ηp^2^ = 0.755; F_2,24_ = 37.073; *p* < 0.001). Figure 1 shows the pairwise comparisons.

#### 3.1.2. Cardiorespiratory Fitness

For cardiorespiratory fitness, the ANOVA analysis revealed a main effect of the moment in the PEDAL group (Wilks’ λ = 0.500; ηp^2^ = 0.500; F_1,44_ = 43.940; *p* < 0.001) but there was no double interaction between the moment and the group (Wilks’ λ = 0.929; ηp^2^ = 0.071; F_1,44_ = 3.370; *p* = 0.073) (see Table 1).

#### 3.1.3. Physical Activity

As can be observed in Table 1, a main effect of moment was discovered for PEDAL PA (Wilks’ λ = 0.850; ηp^2^ = 0.150; F_1,45_ = 7.914; *p* = 0.007) and PEDAL METs (Wilks’ λ = 0.781; ηp^2^ = 0.219; F_1,45_ = 12.593; *p* = 0.001), but not for PEDAL sedentary time (Wilks’ λ = 0.938; ηp^2^ = 0.62; F_1,44_ = 2.915; *p* = 0.095), for the physical activity analysis. In addition, there was a significant interaction effect between the moment and the group in every physical activity aspect (IPAQ PA: Wilks’ λ = 0.908; F_1,45_ = 4.581; *p* = 0.038; ηp^2^ = 0.092; METs: Wilks’ λ = 0.891; F_1,45_ = 5.506; *p* = 0.023; ηp^2^ = 0.109; sedentary time: Wilks’ λ = 0.899; F_1,44_ = 4.916; *p* = 0.032; ηp^2^ = 0.101).

#### 3.1.4. Language Competence

For language competence, no effect of moment was discovered for the PEDAL group (Wilks’ λ = 1.000; ηp^2^ = 0.000; F_1,46_ = 1.000; *p* = 0.943) and no double interaction between moment and group (Wilks’ λ = 0.976; ηp^2^ = 0.024; F_1,46_ = 1.110; *p* = 0.298) was found (see Table 1).

#### 3.1.5. Attention

With respect to the evaluation of attention, the analysis revealed a main effect of moment for the PEDAL group (TOT test: Wilks’ λ = 0.482; ηp^2^ = 0.518; F_1,45_ = 48.293; *p* < 0.001; CON test: Wilks’ λ = 0.388; ηp^2^ = 0.612; F_1,45_ = 71.122; *p* < 0.001). However, there was no double interaction between moment and group (TOT test: Wilks’ λ = 0.972; ηp^2^ = 0.028; F_1,45_ = 1.286; *p* = 0.263; CON test: Wilks’ λ = 0.998; ηp^2^ = 0.002; F_1,45_ = 0.104; *p* = 0.749).

The pairwise comparisons are shown in Table 1.

#### 3.1.6. Heart Rate

Table 2 shows the average HR during the intervention. Comparing the data of the PEDAL pre-test with those of the follow-up weeks showed no effect of any moment for HR_week,_ or HR (PE). However, there was a significant difference between the pre-test HR data (LA) and Week 1 (*p* = 0.001) and Week 7 (*p* = 0.009). Moreover, regarding language, there was a significant interaction effect between the moment and the group for the HR (LA) (Wilks’ λ = 0.174; ηp^2^ = 0.826; F_3,11_ = 17.412; *p* < 0.001), and the PEDAL group HR (LA) was higher than that of the control group during Week 1 (*p* = 0.016).

#### 3.1.7. Additional Results

Regarding gender differences, the IPAQ reported that the physical activity was different from the pre-test (*p* = 0.014) for only the girls from the PEDAL group (see Table 1). Additionally, their HR during language was different both from the HR during the pre-test (Week 1 *p* = 0.003; Week 7 *p* = 0.010) and from the HR of the control group (Week 1 *p* = 0.011; Week 4 *p* = 0.050). These differences were not found in the boys in the PEDAL group (see Table 2). In addition, it can be observed in Table 1 that, during the study, the girls from the PEDAL group reported a higher increase in physical activity (∆ = 1142.41 ± 403.91 MET-min/week) as compared to the boys in the PEDAL group (∆ = 772.93 ± 358.02 MET-min/week).

Finally, within the PEDAL analysis subgroups (splitting groups according to their initial physical activity, METs), there was a significant difference in onset between the physical activity performed by the high-level group and that in the low-level (*p* = 0.003) and moderate-level (*p* < 0.001) groups (see Figure 2). Physical gains from the pre- and the post-intervention in cardiorespiratory fitness were only found for the low-level and moderate-level subgroups (*p* = 0.002, *p* < 0.001, respectively). In regard to physical activity (METs), only that in the low subgroup significantly increased (*p* = 0.007). The high-level subgroup did not improve from the pre-test in any aspect beyond reducing their sedentary time (*p* = 0.049). Regarding academic performance and attention, non-remarkable differences were found. Figure 2 shows the evolution of every physical variable according to the students’ previous physical activity (low, moderate, or high).

## 4. Discussion

It is crucial to increase children and adolescents’ physical activity to ameliorate the pandemic-induced inactivity and sedentarism, especially in certain groups of the population. The implementation of bike desks positively influenced students’ overall physical activity and reduced their sedentary time without compromising their academic performance. Therefore, it could be considered a good strategy to implement in schools.

Regarding physical condition, both the PEDAL and the control group significantly improved their cardiorespiratory fitness but no significant differences between the groups were found at the end of the intervention. Since both groups improved their physical condition, it seemed to be due to the restart of physical education lessons and other active routines after the summer, during the first few months of the school year. Based on previous studies [10,27], we would have expected, after ten weeks of cycling and a physical activity increase, a larger improvement in the PEDAL group than the control group, but it is possible that the PEDAL light-moderate intensity workout and/or the project length were not sufficient.

The previously mentioned benefits in the reported physical activity were in favor of the PEDAL group. Only this group increased their physical activity levels and energy expenditure from the pre-test. Moreover, after the intervention, the PEDAL group registered significantly less sedentary time and more physical activity than the control group. Previous studies with active desks are in line with this result; they recorded less sedentary time for the intervention group during academic lessons [28] and in workplaces [29], as well as higher physical activity when cycling on a bike desk [10,13].

According to the HR data analysis, during school-time, the statistics disclosed different intervention moments during active language lessons, where the HR was significantly higher in the PEDAL group than in the control group and higher than pre-intervention data from traditional lessons. The HR increase due to physical activity is positive in combating the health risks of a sedentary lifestyle, although the bike desks did not manage to achieve the MVPA, the recommended physical activity for children and adolescents’ health [30]. Thus, analyzing the HR monitors’ data, the PE lessons were the only moments at which the participants achieved a higher HR and sustained MVPA, while bike desks contributed to low to moderate physical activity. This fact confirms the inestimable value that PE has in achieving the minimum amount of time engaged in moderate to vigorous physical activity [31,32]. Although students could not achieve the MVPA, the light physical activity performed with the bike desks should be seen as valuable in school-based physical activity interventions [33]. Simultaneously engaging in low-intensity exercise and paying attention to the lesson could be fundamental to maintain the academic dynamics of the language lesson and to not interfere with the learning process. Previous studies presented different and opposing physical activity intensities when working out on a bike desk; on the one hand, an investigation did obtain a greater MVPA for the intervention group but did not examine whether this could cause negative academic effects [12]; on the other hand, another work stated that students only engaged in light physical activity [17].

In terms of academic performance, there were no effects on language competence, attention, and concentration since these variables were significantly better in both groups after the intervention, with no post-test differences between groups. This finding of no academic detriment agrees with earlier active desk studies [34] and bike desk interventions [35]. Moreover, a systematic review about lessons using active desks did find positive effects on academic motivation, attention, and on-task behavior [11].

Furthermore, this is the first study disentangling the possible effects of bike desks in terms of gender and physical activity baseline levels. The lack of data collection regarding gender was a recurring limitation in previous bike desk studies in education [17]. The hypothesis is confirmed because, in several aspects, girls obtained better results than boys, which leads us to believe that the PEDAL program could have contributed to a more profound motivation for physical activity in girls. In a classroom-based physical intervention [36], better results were also obtained for girls, and they explained this result considering that the program exerted relatively less influence on the boys’ overall physical activity levels since boys usually have a higher physical activity level than girls [37].

Finally, the analysis appeared to confirm the hypothesis that adolescents with low activity levels could benefit more from the PEDAL intervention. In the same vein, several school interventions have found significant increases in step counts among children with low activity levels [38,39].

Like all research, this investigation has strengths as well as some limitations that must be taken into account when interpreting the results. Future investigations, with larger sample sizes, are necessary to analyze intervention effects in specific groups, and a more extended program could have revealed a wider range of benefits after the intervention. Moreover, the incorporation of different locations could allow for generalizability. Furthermore, the language competence test was not validated and some biometric variables, such as body mass and BMI, were not recorded, but both aspects could have conditioned the results obtained.

Taking all of the limitations into account, future lines of research could check students’ long-term adherence to the PEDAL program and they could explore the effects according to the intensity of the physical activity performed using bike desks.

In conclusion, the presented bike desk educational intervention is a feasible way to increase students’ daily light–moderate physical activity during school time, with no negative academic interference. A minor trend, which needs to be studied in depth, suggests that the intervention could impact certain subgroups of the population, such as girls and adolescents with low to moderate physical activity levels, who are usually difficult to reach.

## Figures and Tables

**Figure 1 children-10-01641-f001:**
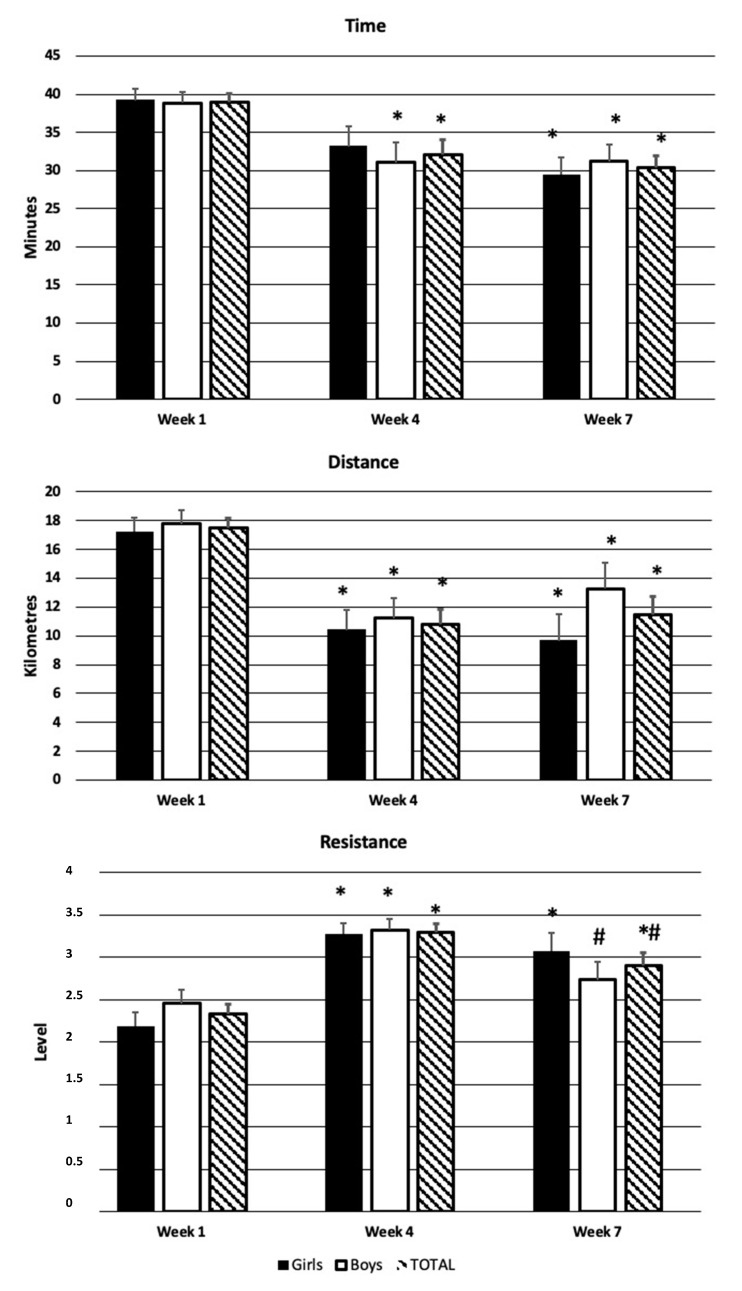
Bike desk records. Data presented as mean ± SEM. * *p* < 0.05 different from week 1; # *p* < 0.05 different from week 4.

**Figure 2 children-10-01641-f002:**
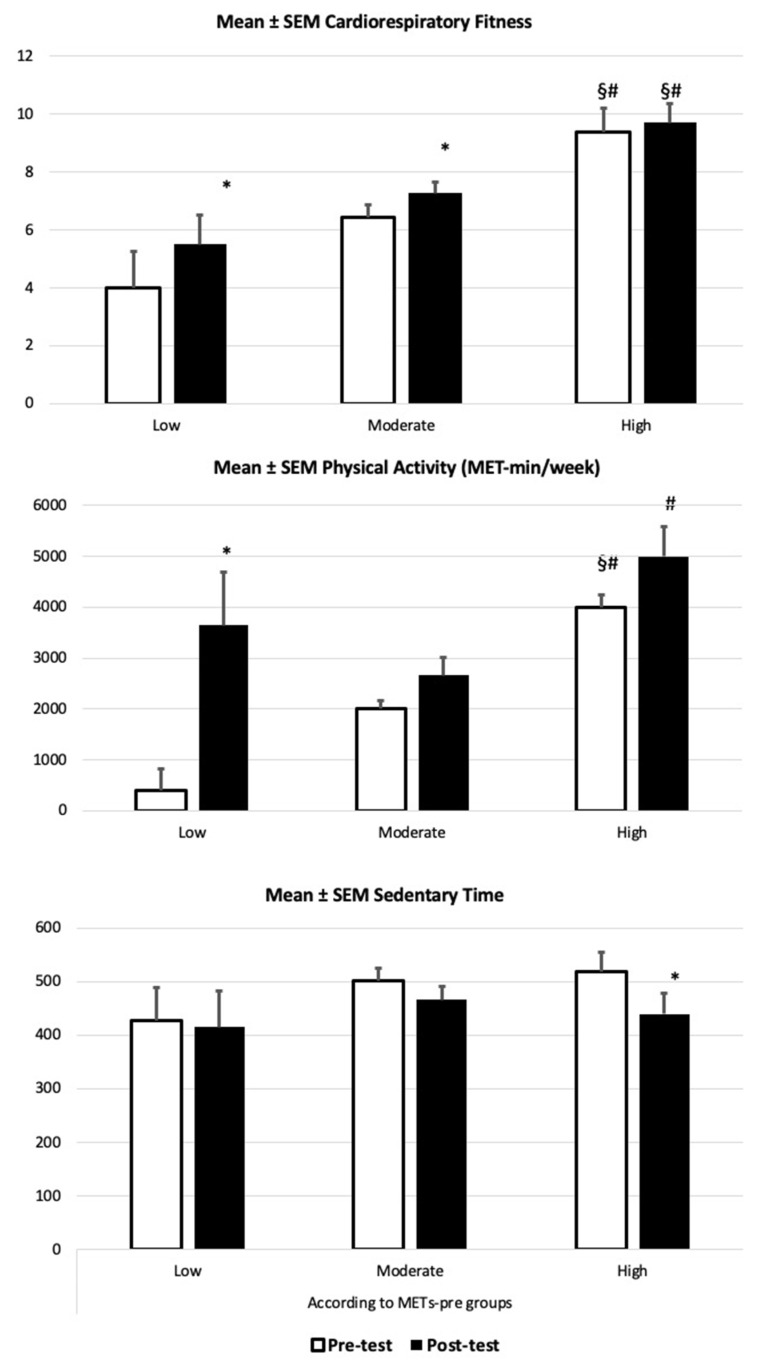
Evolution of the physical variables according to the students’ physical activity and physical fitness groups (low, moderate, or high). Data presented as mean ± SEM. * *p* < 0.05 different from low group; # *p* < 0.05 different from moderate group; § *p* < 0.05 different from pre-test.

**Table 1 children-10-01641-t001:** Pre- and post-intervention participant characteristics. Data presented as mean ± SEM.

		PEDAL Group			Control Group		
		Girls	Boys	Total	Girls	Boys	Total
CF pre		4.23 ± 0.48	8.62 ^§^ ± 0.48	6.42 ± 0.34	4.64 ± 0.52	8.59 ^§^ ± 0.52	6.61 ± 0.37
CF post		5.42 * ± 0.42	9.19 *^§^ ± 0.42	7.30 * ± 0.30	5.50 * ± 0.46	8.77 ^§^ ± 0.46	7.14 * ± 0.32
IPAQ-SF	METs pre	1683.27 ± 293.00	2898.47 ^§^ ± 259.72	2290.87 ± 195.77	2072.21 ± 280.53	2539.75 ± 280.53	2305.98 ± 198.36
	METs post	2825.68 * ± 406.53	3671.40 * ± 360.35	3248.54 *^#^ ± 271.62	1737.83 ± 389.22	2925.08 ^§^ ± 389.22	2331.46 ± 275.22
	Sedentary time pre	505.91 ± 36.00	499.29 ± 31.91	502.60 ± 24.06	501.25 ± 34.47	490.91 ± 36.00	496.08 ± 24.92
	Sedentary time post	468.18 ± 30.02	447.86 ^#^ ± 26.61	458.02 ^#^ ± 20.06	522.50 ± 28.74	545.46 ± 30.02	533.98 ± 20.78
Language competence pre		6.20 ± 0.43	5.37 ± 0.41	5.78 ± 0.30	6.29 ± 0.45	5.40 ± 0.47	5.84 ± 0.32
Language competence post		6.37 ± 0.34	5.23 ^§^ ± 0.33	5.80 ± 0.24	5.50 * ± 0.36	5.60 ± 0.37	5.55 ± 0.26
Attention	TOT pre	439.50 ± 19.23	405.36 ± 17.80	422.43 ± 13.10	398.00 ± 17.80	388.67 ± 22.21	393.33 ± 14.23
	TOT post	487.25 * ± 22.71	451.71 * ± 21.03	469.48 * ± 15.48	460.71 * ± 21.03	442.00 * ± 26.23	451.36 * ± 16.81
	CON pre	171.67 ± 7.89	157.71 ± 7.30	164.69 ± 5.38	151.43 ± 7.30	148.11 ± 9.11	149.77 ± 5.84
	CON post	202.50 * ± 10.62	183.29 * ± 9.83	192.89 * ± 7.23	184.64 * ± 9.83	174.33 * ± 12.26	179.49 * ± 7.86

Notes: CF: cardiorespiratory fitness; TOT: attention; CON: concentration. * *p* < 0.05 different from pre; ^#^ *p* < 0.05 different from the control group; ^§^ *p* < 0.05 different from girls.

**Table 2 children-10-01641-t002:** Evolution of heart rate throughout the intervention. Data presented as mean ± SEM.

		PEDAL Group				Control Group			
		Pre-Intervention	Week 1	Week 4	Week 7	Pre-Intervention	Week 1	Week 4	Week 7
HR_week_	Girls	90.92 ± 4.47	87.96 ± 4.65	92.62 ± 4.92	96.63 ± 4.64	90.65 ± 5.00	84.95 ± 5.20	87.40 ± 5.50	89.69 ± 5.19
	Boys	93.70 ± 5.00	88.12 ± 5.20	98.28 ± 5.50	97.00 ± 5.19	94.30 ± 5.00	91.25 ± 5.20	98.98 ± 5.50	99.00 ± 5.20
	Total	92.31 ± 3.35	88.04 ± 3.49	95.45 ± 3.69	96.82 ^b^ ± 3.48	92.48 ± 3.53	88.10 ± 3.68	93.19 ± 3.89	94.34 ± 3.67
HR (LA lessons)	Girls	86.90 ± 4.68	98.25 ^a#^ ± 4.27	94.12 ^#^ ± 5.51	99.07 ^a^ ± 5.10	84.38 ± 5.24	79.25 ± 4.77	76.29 ± 6.16	84.29 ± 5.70
	Boys	88.44 ± 5.24	96.23 ± 4.77	93.78 ± 6.16	94.94 ± 5.70	88.69 ± 5.24	89.56 ± 4.77	90.83 ± 6.16	91.46 ± 5.70
	Total	87.67 ± 3.51	97.24 ^a#^ ± 3.20	93.95 ± 4.13	97.00 ^a^ ± 3.82	86.53 ± 3.70	84.41 ± 3.37	83.56 ± 4.35	87.87 ± 4.03
HR (PE lessons)	Girls	129.30 ± 4.43	122.30 ± 4.53	141.30 ^b^ ± 5.28	123.10 ± 4.33	121.25 ± 4.95	119.88 ± 5.07	127.25 ± 5.91	127.25 ± 4.84
	Boys	132.13 ± 4.95	119.88 ± 5.07	129.38 ± 5.91	130.38 ± 4.84	143.50 ^§^ ± 4.95	124.50 ^a^ ± 5.07	128.25 ± 5.91	129.75 ± 4.84
	Total	130.71 ± 3.32	121.09 ± 3.40	135.34 ^b^ ± 3.96	126.74 ± 3.25	132.38 ± 3.50	122.19 ± 3.58	127.75 ± 4.18	128.50 ± 3.42

Notes: HR: heart rate; LA: language; PE: physical education. ^a^ *p* < 0.05 different from pre-intervention; ^b^ *p* < 0.05 different from week 1; ^#^ *p* < 0.05 different from the control group; ^§^ *p* < 0.05 different from girls.

## Data Availability

Data are unavailable due to privacy or ethical restrictions.

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
