# Peer review of "Do You Want to Increase Physical Activity in Adolescents? A School-Based Physical Activity Program Could Be an Efficient Way"

_children, 2023, doi:10.3390/children10101641_

Round 1

Reviewer 1 Report

Thank you for the opportunity to review this article!

I want to appreciate the skills of the authors to design the research design, but also to concisely present the statistical information obtained, clearly, concisely and coherently.

There are some aspects, more on the technical side than the content, that should be corrected, so that the article can be published.

- I suggest a more concise and targeted title.

- I suggest identifying some keywords to send more targeted content (perhaps related to tools, type of activity, etc.)

- I suggest the introduction of some information on the tools used in the Abstract.

- there is a large number of sources in the text that are not found in the bibliography (for example, Abarca-Gomez, Kriemler et al., 2011, Cecchini &, 2020, MAvilidi & , 2020, Torbeyns et al., 2014, Josaphat et al ., 2019, Mueller et al., 2017, Pontifex et al., 2013, Donnelly & Lambourne, 2011, Bakeret al., 2019, Sherry &, 2017, Bailey &, 2012, Kang & Brinhampt, 2019)

- correction of inconsistencies between the years and authors in the text and those written in the References (for example, WHO, 2013 is in the bibliography, but in the text, on line 41, it is WHO, 2010);

- line 55 – Torbeyns & al., 2014, and 2017 appears in References

- row 59, 63 – to which Fedewa et al., 2017 refers? 2017a and 2017b should be differentiated

- lines 114,115, 141 – is the year for the Brickenkamp source 1998? Or 2002

- lines 114 and 139 – Resolution of 8 of ... should be entered at References

- lines 74-82 - must be deleted - are information about the standards of the journal

- compliance with the standards of the journal regarding references to sources in the text, numbering of sub-chapters

- to take into account confusing forms that refer to the physical activities performed in other topics and the physical education lessons. physical education lessons have clear objectives and contents. pedaling activities (independent variable?) were included as physical activities during other lessons and aimed to identify the effects on the subjects' physical and academic performance. it is not clearly specified if during certain tasks for the respective topics, the subjects were pedaling and also had specific tasks for that topic.

- to specify clearly which are the dependent and independent variables

- to specify if the issued hypotheses are confirmed/denied

- to identify and include the limits of the study.

Author Response

Dear Editor,

We really appreciate all the comments and suggestions of all the reviewers and the possibility to review the manuscript. We are sure that all these changes will improve the quality of the work. Below you will find the author’s comments to each of the aspects pointed out by the reviewers.

Comments:

Reviewer 1

Thank you for the opportunity to review this article!

I want to appreciate the skills of the authors to design the research design, but also to concisely present the statistical information obtained, clearly, concisely and coherently.

There are some aspects, more on the technical side than the content, that should be corrected, so that the article can be published.

Dear reviewer,

Thank you very much for giving us the opportunity to review the document. We greatly appreciate your comments and suggestions to improve the content of the article and make it easier for future readers to understand. Below are the changes we have made:

I suggest a more concise and targeted title.

Your comment is greatly appreciated. We have modified the title to a more concise and specific one, with the aim of making it more attractive to read by future read.

I suggest identifying some keywords to send more targeted content (perhaps related to tools, type of activity, etc.)

The keywords have been revised to better adapt them to the content of the manuscript.

I suggest the introduction of some information on the tools used in the Abstract.

The tools have been introduced in the abstract, thank you very much for pointing it out.

There is a large number of sources in the text that are not found in the bibliography (for example, Abarca-Gomez, Kriemler et al., 2011, Cecchini &, 2020, MAvilidi & , 2020, Torbeyns et al., 2014, Josaphat et al ., 2019, Mueller et al., 2017, Pontifex et al., 2013, Donnelly & Lambourne, 2011, Bakeret al., 2019, Sherry &, 2017, Bailey &, 2012, Kang & Brinhampt, 2019). Correction of inconsistencies between the years and authors in the text and those written in the References (for example, WHO, 2013 is in the bibliography, but in the text, on line 41, it is WHO, 2010);

Line 55 – Torbeyns & al., 2014, and 2017 appears in References

Row 59, 63 – to which Fedewa et al., 2017 refers? 2017a and 2017b should be differentiated

Lines 114,115, 141 – is the year for the Brickenkamp source 1998? Or 2002

Lines 114 and 139 – Resolution of 8 of ... should be entered at References

Thank you very much for your corrections, we are ashamed of the mistakes. The bibliography has been revised in depth. The lack of information and mistakes have been corrected.

Lines 74-82 - must be deleted - are information about the standards of the journal

Compliance with the standards of the journal regarding references to sources in the text, numbering of sub-chapters

Thank you very much for your observation. That paragraph should have been eliminated prior to sending the document to the journal. It has already been removed in this revision.

To take into account confusing forms that refer to the physical activities performed in other topics and the physical education lessons. physical education lessons have clear objectives and contents. pedaling activities (independent variable?) were included as physical activities during other lessons and aimed to identify the effects on the subjects' physical and academic performance. it is not clearly specified if during certain tasks for the respective topics, the subjects were pedaling and also had specific tasks for that topic.

We have tried to clarify it in the procedure "Neither PEDAL nor control group didn´t change their usual language arts lesson dynamics and tasks, the only difference resided in voluntary pedaling for PEDAL group students during the aforementioned lessons".

To specify clearly which are the dependent and independent variables

Following your suggestion, we have specified in the data analysis section which are the dependent variables, and which are the independent variables.

To specify if the issued hypotheses are confirmed/denied

Following your suggestion, we have included some statements in the discussion to confirm or deny the hypotheses.

To identify and include the limits of the study.

Thank you very much for your suggestion. The limits of the study have been added in the discussion.

Thank you very much for your hard work and your help with our paper.

Reviewer 2 Report

I appreciate the opportunity to review the paper title “ Could a school-based physical activity program be an efficient way to increase physical activity in adolescents? “ submitted to children. This is an important topic and I hope the authors find my comments helpful in further strengthening and improving the manuscript. 

1.The abstract despite its clarity in detailing the experiment's procedures, setup, and findings, has some notable deficiencies.

(1)The aim of the study is not precisely defined. Detailing the expected degree of impact on academic and physical performance can contribute to a firmer measurement of outcomes.

(2) The abstract does not offer enough contextual information on the Program to Enhance and Develop Active Lessons (PE-DAL) – how it functions, what the "bike desks" are, and what kind of activity it involves.

(3)The study uses measures such as TOT and CON, but does not specify what these terms mean. This makes it difficult for readers to fully understand the methods and results.

(4)While the improvement in physical activity is identified in the PEDAL group, the research shows no significant improvement in academic performance nor cardiorespiratory fitness. The contradiction requires explanation, and hypothesis reevaluation may be needed. 

2.The following deficiencies can be observed in the Introduction:

(1) The introduction does begin with a broad context about the global phenomena of increasing child obesity and sedentarism. However, the transition from these broad problems to the specific problem that the study is addressing (the effects of the PEDAL program) could be made more clearly and gradually.

(2)Even though the introduction provides a good backdrop on the current state of children's inactivity and obesity, it lacks a clear theoretical framework or concepts underpinning the PEDAL program and bike desks.

(3)While the research gap has been identified, it needs to be further emphasized as to why studying the impact of bike desks on academic performance is crucial.

(4)The introduction does not explicitly state why the research is important or the potential implications of its findings.

(5) Although previous research is mentioned, a critical analysis or comparison of findings from past research on the same topic is missing.

(6)The specific aims of PEDAL should be expressly stated.

(7) There is some use of terminology (e.g., active recess, bike desks, physically active learning) without clear definitions. This could confuse readers not familiar with the specific field.

(8) Although the aim of the research is mentioned towards the end, it is not clearly framed as a novel addition to the field. It is vital to explicitly state the unique contribution of the research.

(9)Remove the final paragraph from the introduction.

3.Several deficiencies are observed in the methods section.

(1)Sampling:The sample size is small (55 final participants), which may limit the power and generalizability of the results. No sample size calculation or rationale for selecting 60 schoolers is provided.

(2)The use of self-reported measurement tools like IPAQ and pedaling log may introduce bias as students can over or under-report their activities. How their accuracy was ruled-out is unspecified.

(3) The specifics of the intervention group's activities are not adequately discussed, particularly how the active desk and pedaling were incorporated into the normal routine.

(4) The activities performed by the control group during language arts period are not clearly detailed. It's important to know what was done to negate placebo effects.

(5)The timing of the testing is not clear – what was the specific time period between pre and post-tests for both groups? This can greatly impact understanding of the intervention's effects over time.

(6)The study is based on a single location (a public high school in Madrid), which limits its generalizability.

(7) It's unknown whether the language competence test was validated and reliable for this age group and context. Because regional tests may not align with academic standards elsewhere, potential biases could be present.

4.The results have the following deficiencies:

(1)Lack of Clarity and Detail: While many of the results were statistically significant, the explanation and contextualization of the findings are quite poor. The practical implications of the differences observed are not well drawn out.

(2)Sample Homogeneity: Insights regarding how the biometric variables of the participants (like age, height, body mass, BMI, etc.) may have influenced the results are missing.

(3) To give a complete picture of the results, all findings, significant or non-significant, should be reported. For example, more detail should be given about the lack of a "double interaction" in the findings.

(4)The classification logic for the low, moderate, and high physical activity subgroups is not clearly explained. Did these labels come from the IPAQ result?

5.The discussion presents the following deficiencies:

(1)While the effects of the intervention are properly discussed, the discussion lacks critique and depth, particularly, in reflecting upon the aspects where the results did not align with the hypotheses.

(2)Direct Comparisons: The comparison between the PEDAL and control groups is often implied, but not explicitly stated. Dedicated comparisons deepen understanding of the study's outcomes.

(3)Although the improvements are highlighted, the discussion is lacking in a thorough address of the limitations.

(4)Assertions about the value the PEDAL programme could have for low-activity level adolescents and girls are based on trends, and not firmly based on data. This could lead to misleading conclusions.

(5)Suggestions for future investigation are dispersed throughout the discussion. These could instead be consolidated towards the end of the discussion.

Author Response

Dear Editor,
We really appreciate all the comments and suggestions of all the reviewers and the possibility to review the manuscript. We are sure that all these changes will improve the quality of the work. Below you will find the author’s comments to each of the aspects pointed out by the reviewers.

Comments:
Reviewer 2
I appreciate the opportunity to review the paper title “Could a school-based physical activity program be an efficient way to increase physical activity in adolescents? “submitted to children. This is an important topic and I hope the authors find my comments helpful in further strengthening and improving the manuscript. 
Dear reviewer,
Thank you very much for giving us the opportunity to review the document. We greatly appreciate your comments and suggestions to improve the content of the article and make it easier for future readers to understand.  Below are the changes we have made:

Abstract
The abstract despite its clarity in detailing the experiment's procedures, setup, and findings, has some notable deficiencies.
The aim of the study is not precisely defined. Detailing the expected degree of impact on academic and physical performance can contribute to a firmer measurement of outcomes.
Thank you very much for your comment. Information has been added to the sentence to clearly define the objective of the study.

The abstract does not offer enough contextual information on the Program to Enhance and Develop Active Lessons (PE-DAL) – how it functions, what the "bike desks" are, and what kind of activity it involves.
Thanks for your advice, the abstract now says "This intervention based on pedal or bike desks, stationary bikes that integrate with a desk workspace" and "the PEDAL students pedaled 4 days a week for 10 weeks, during their Spanish language arts lessons"

The study uses measures such as TOT and CON, but does not specify what these terms mean. This makes it difficult for readers to fully understand the methods and results.
Following your instructions, we have defined the terms TOT and CON to improve the understanding of the text while reading it.

While the improvement in physical activity is identified in the PEDAL group, the research shows no significant improvement in academic performance nor cardiorespiratory fitness. The contradiction requires explanation, and hypothesis reevaluation may be needed. 
Thanks for highlighting it. We have explained it better in the abstract.
Besides, in the discussion we explain that "Based on previous studies (Torbeyns et al. 2017; Polo-Recuero, Moreno-Barrio, and Ordóñez-Dios 2020), it would have been expected, after ten weeks of cycling and a physical activity increase, a higher improvement of the PEDAL group than the control group but maybe the PEDAL work out light intensity and/or the project length were not enough".

Introduction
The following deficiencies can be observed in the Introduction:
The introduction does begin with a broad context about the global phenomena of increasing child obesity and sedentarism. However, the transition from these broad problems to the specific problem that the study is addressing (the effects of the PEDAL program) could be made more clearly and gradually.
Thank you very much for your observation, we have included a sentence that could explain better why schools reach every adolescent and are important to achieve the main objective.

Even though the introduction provides a good backdrop on the current state of children's inactivity and obesity, it lacks a clear theoretical framework or concepts underpinning the PEDAL program and bike desks.
Thanks to your advice we noticed that we could give more information about different active desks to properly introduce bike desks and understand the topic.
While the research gap has been identified, it needs to be further emphasized as to why studying the impact of bike desks on academic performance is crucial.
A new sentence has been included, detailing why it is important to keep studying PAL strategies, bike desk in particular, because even though they increase physical activity they could be a difficulty for students´ learning. 

The introduction does not explicitly state why the research is important or the potential implications of its findings.
Thank you very much, now we have finished the introduction explaining that it seems that PEDAL program could help to be closer to the physical activity goal set by the WHO and we analyze in depth in the discussion the potential implications of its findings.

Although previous research is mentioned, a critical analysis or comparison of findings from past research on the same topic is missing.
Thanks to your advice we noticed that we could mention the systematic review we did analyzing the background of bike desks and add more information.
Polo-Recuero, B., Rojo-Tirado, M. Á., Ordóñez-Dios, A., Breitkreuz, D., & Lorenzo, A. (2021). The Effects of Bike Desks in Formal Education Classroom-Based Physical Activity: A Systematic Review. Sustainability, 13(13), 7326.

The specific aims of PEDAL should be expressly stated.
Thank you very much for your comment. The aims have been rewritten.

There is some use of terminology (e.g., active recess, bike desks, physically active learning) without clear definitions. This could confuse readers not familiar with the specific field.
The requested definitions have been included.

Although the aim of the research is mentioned towards the end, it is not clearly framed as a novel addition to the field. It is vital to explicitly state the unique contribution of the research.
Thank your very much for your comment, we agree that it was crucial to highlight why PEDAL is important (PEDAL is a pioneer establishing bike desks in a school in Spain and PEDAL studies the different gender response to the strategy).

Remove the final paragraph from the introduction.
Thank you very much for your observation. That paragraph should have been eliminated prior to sending the document to the journal. It has already been removed in this revision.

Methods
Several deficiencies are observed in the methods section.
Sampling:The sample size is small (55 final participants), which may limit the power and generalizability of the results. No sample size calculation or rationale for selecting 60 schoolers is provided.
Thank you very much for pushing us to explain it. We have now justified why the sample had that size and we compare it with the sample of other experimental studies using bike desks.

The use of self-reported measurement tools like IPAQ and pedaling log may introduce bias as students can over or under-report their activities. How their accuracy was ruled-out is unspecified.
Thank you very much, we have added that prior to the implementation of the PEDAL program, during the 2018-2019 school year, a pilot project was carried out in order to help students to become familiar with some of the instruments.
Besides, we reference the investigation of Craig (2003) that comfirms the reliability and validity of IPAQ. In spanish we have Martínez-Gómez et al. (2009) research:
Martínez-Gómez, D., Martínez-de-Haro, V., Pozo, T., Welk, G. J., Villagra, A., Calle, M. E., ... & Veiga, O. L. (2009). Fiabilidad y validez del cuestionario de actividad física PAQ-A en adolescentes españoles. Revista española de salud pública, 83, 427-439.

The specifics of the intervention group's activities are not adequately discussed, particularly how the active desk and pedaling were incorporated into the normal routine.
The activities performed by the control group during language arts period are not clearly detailed. It's important to know what was done to negate placebo effects.
Thank you very much for your comment, the procedure has been rewritten to incorporate more information in a paragraph about voluntary pedaling for PEDAL group students during the Language arts lessons normal tasks.

The timing of the testing is not clear – what was the specific time period between pre and post-tests for both groups? This can greatly impact understanding of the intervention's effects over time.
Thank you very much for your constructive comment. More information on how was the timing has been added.

The study is based on a single location (a public high school in Madrid), which limits its generalizability.
Thank you very much for pointing it out. It has been added as a limit of the study in the discussion.

It's unknown whether the language competence test was validated and reliable for this age group and context. Because regional tests may not align with academic standards elsewhere, potential biases could be present.
Thank you very much for your contribution to strengthen the methods. The language test is widely used in Comunidad de Madrid (every highschool apply this test) but, as you highlight it is not validated and reliable. We have pointed it out in the limitations of the study.

Results
The results have the following deficiencies:
Lack of Clarity and Detail: While many of the results were statistically significant, the explanation and contextualization of the findings are quite poor. The practical implications of the differences observed are not well drawn out.
Thank you very much, we have tried to contextualize it in depth in our conclusions and discussion, adding details that could explain the results and including some sentences that could give the practical implications.

Sample Homogeneity: Insights regarding how the biometric variables of the participants (like age, height, body mass, BMI, etc.) may have influenced the results are missing.

Thank you for your comment. We have included in our limits of the study that some biometric variables were not recorded, such as body mass or BMI, which could condition the results obtained.

To give a complete picture of the results, all findings, significant or non-significant, should be reported. For example, more detail should be given about the lack of a "double interaction" in the findings.
We really appreciate your suggestion to strengthen the results section. To improve understanding of the results set, the section has been rewritten, linking each paragraph to the table/figure where the results can be consulted.

The classification logic for the low, moderate, and high physical activity subgroups is not clearly explained. Did these labels come from the IPAQ result?
You are right, we have included a short sentence in the instruments to clarify it: According to IPAQ-SF, results are shown as grades of physical activity performed (1: low, 2: moderate, 3: high), metabolic equivalents (MET-minute/week) and sedentary time (minutes/week)...

Discussion
The discussion presents the following deficiencies:
While the effects of the intervention are properly discussed, the discussion lacks critique and depth, particularly, in reflecting upon the aspects where the results did not align with the hypotheses.
Thank you very much for your contribution to strengthen the discussion of the work. We have now tried to go deeper and to justifiy the hipotheses that are denied. We explain that the intensity, duration, sample... could have been the main cause of the unexpected results.

Direct Comparisons: The comparison between the PEDAL and control groups is often implied, but not explicitly stated. Dedicated comparisons deepen understanding of the study's outcomes.
Thanks for pointing it out, we have incorporated the acceptance or denial of the hypotheses in the discussion to make the comparisons easier to understand.

Although the improvements are highlighted, the discussion is lacking in a thorough address of the limitations.

Thank you very much for your suggestion. The limits of the study have been added in the discussion.

Assertions about the value the PEDAL programme could have for low-activity level adolescents and girls are based on trends, and not firmly based on data. This could lead to misleading conclusions.

Thank you very much for your comment, we have cautiously clarified that it seems to be a trend that confirms the hypothesis that low activity level adolescents could have benefited more from PEDAL intervention but future investigations, with larger samples, are necessary to analyze intervention effects in specific groups.

Suggestions for future investigation are dispersed throughout the discussion. These could instead be consolidated towards the end of the discussion.
Following your request, we have added a few ideas on future research and some cautious recommendations, regarding the study group.

Thank you very much for your hard work and your help with our paper.

Round 2

Reviewer 2 Report

The text has been thoroughly revised, incorporating the feedback provided by the reviewer. The authors have successfully addressed my comments and suggestions, enhancing both the quality and clarity of the manuscript. I am now in a position to recommend its acceptance.